# Possible Strategies to Reduce the Tumorigenic Risk of Reprogrammed Normal and Cancer Cells

**DOI:** 10.3390/ijms25105177

**Published:** 2024-05-09

**Authors:** Ying-Chu Lin, Cha-Chien Ku, Kenly Wuputra, Chung-Jung Liu, Deng-Chyang Wu, Maki Satou, Yukio Mitsui, Shigeo Saito, Kazunari K. Yokoyama

**Affiliations:** 1School of Dentistry, Kaohsiung Medical University, Kaohsiung 80708, Taiwan; chulin@cc.kmu.edu.tw; 2Graduate Institute of Medicine, Department of Medicine, Kaohsiung Medical University, Kaohsiung 80708, Taiwan; r991046@gap.kmu.edu.tw (C.-C.K.); kenlywu@hotmail.com (K.W.); 3Regenerative Medicine and Cell Research Center, Kaohsiung Medical University, Kaohsiung 80708, Taiwan; pinkporkkimo@yahoo.com.tw (C.-J.L.); dechwu@yahoo.com (D.-C.W.); 4Cell Therapy and Research Center, Kaohsiung Medical University Hospital, Kaohsiung 80756, Taiwan; 5Waseda Research Institute for Science and Engineering, Waseda University, Tokyo 169-8555, Japan; 6Division of Gastroenterology, Department of Internal Medicine, Kaohsiung Medical University Hospital, Kaohsiung 80756, Taiwan; 7Research Institute, Horus Co., Ltd., Iruma 358-0032, Saitama, Japan; satou.maki@horus-jnl.co.jp (M.S.); mitsui.yukio@horus-jnl.co.jp (Y.M.); 8Saito Laboratory of Cell Technology, Yaita 329-1571, Tochigi, Japan

**Keywords:** induced pluripotent stem cells, organoids, regenerative medicine, reprogramming, therapeutic application, tumorigenic risk

## Abstract

The reprogramming of somatic cells to pluripotent stem cells has immense potential for use in regenerating or redeveloping tissues for transplantation, and the future application of this method is one of the most important research topics in regenerative medicine. These cells are generated from normal cells, adult stem cells, or neoplastic cancer cells. They express embryonic stem cell markers, such as OCT4, SOX2, and NANOG, and can differentiate into all tissue types in adults, both in vitro and in vivo. However, tumorigenicity, immunogenicity, and heterogeneity of cell populations may hamper the use of this method in medical therapeutics. The risk of cancer formation is dependent on mutations of these stemness genes during the transformation of pluripotent stem cells to cancer cells and on the alteration of the microenvironments of stem cell niches at genetic and epigenetic levels. Recent reports have shown that the generation of induced pluripotent stem cells (iPSCs) derived from human fibroblasts could be induced using chemicals, which is a safe, easy, and clinical-grade manufacturing strategy for modifying the cell fate of human cells required for regeneration therapies. This strategy is one of the future routes for the clinical application of reprogramming therapy. Therefore, this review highlights the recent progress in research focused on decreasing the tumorigenic risk of iPSCs or iPSC-derived organoids and increasing the safety of iPSC cell preparation and their application for therapeutic benefits.

## 1. Introduction

Self-renewal and pluripotency, that is, the ability to develop into all cell types other than extraembryonic tissue cells, are two representative features of human induced pluripotent stem cells (iPSCs). Four factors, such as octamer binding protein 4 (OCT4), sex-determining region Y-box 2 (SOX2), Krüppel-like factor 4 (KLF4), and cellular myelocytomatosis oncogene (C-MYC) (4F=OSKM), and different combinations including additional stemness genes like homeobox transcription factor NANOG, were used to reprogram normal human fibroblast cells to generate induced pluripotent stem cells (iPSCs) as described elsewhere [1,2]. Recently, an in vivo study including epigenetic rejuvenation strategies via partial reprogramming in both mouse and human models was described [3].

One major problem with the clinical application of iPSCs is the possible risk of developing cancer in transplanted cells [1,2], and reprogramming strategies are currently under scrutiny due to their strong impact on safety and efficacy in clinical applications. These strategies can be classified into two major groups, namely integrating and nonintegrating gene transfer systems. The former involves the integration of the recombinant vector DNA into the host genome. Among such systems, the application strategies of viral RNA vectors (retrovirus and lentivirus) differ from those of nonviral vectors (linear DNA and transposons) [3,4,5]. Nonintegrating viral vectors like adenovirus and Sendai virus and nonintegrating vectors (episomal vectors, RNA, peptides, proteins, and chemicals) can also be used to transfer the reprogramming materials into the host cells.

A challenge in using integrating vectors is the potential lack of silencing the expression of integrated transgenes, which can prevent cell-autonomous maintenance of pluripotency or impact differentiation due to the silenced expression of transgenes or their re-expression during differentiation, thereby possibly leading to the generation of tumorigenic cells [6].

The reprogramming efficiency of nonintegrating strategies to derive iPSCs is rather low (approximately 0.001%) [3,7]. Reprogramming of cells can also be performed via chemical stimulation using small molecules that can promote the establishment of animal and human iPSCs [1,8,9,10,11,12]. Moreover, chemical reprogramming is recognized as a promising technique for applying iPSCs in regenerative medicine [1].

In some cases, abnormal or deleted p53 significantly increased not only the reprogramming efficiency of iPSC-like cells (referred to as induced pluripotent cancer stem cells, iPCSCs) but also the tumorigenicity of iPCSCs, as observed in mice derived from iPSCs with *p53* knockout [13,14]. Furthermore, suppression of the expression of anti-oncogenes was observed to not only enhance the reprogramming efficiency of cells but also adversely increase the risk of tumorigenesis after transplantation [15] (Figure 1). These findings, together with many other similar findings, strongly suggest that tumor reprogramming and iPSC generation share similar pathways. Therefore, the risk of tumorigenesis in iPSC-based stem cell applications is a major concern. We speculate that cancer stem cells may be generated through reprogramming.

Reprogramming strategies have been used for several types of cancer cells as a possible means of repressing tumorigenesis [16,17,18]. In general, the reprogramming efficiency of cancer cells is lower than that of somatic cells [19,20]. The epigenetic memory of the original cells is important for reprogramming. It can lead to incomplete reprogramming caused by a failure to reset the epigenesis to an embryonic stem cell (ESC)-like state [21].

The reprogramming efficiency is lower in cancer cells than in normal cells because the cancer epigenome and chromosomal changes or gene mutations are present in cancer cells.

Moreover, incomplete and inefficient resetting of the cancer-associated epigenome could possibly interfere with successful reprogramming. Recently, tumorigenic signals were demonstrated to impair the transcriptional response to the reprogramming factors OCT4, SOX2, KLF4, and C-MYC (OSKM) in clear cell sarcomas [20]. Ito et al. suggested that resistance to reprogramming is a general feature of cancer cells [19].

Moreover, the enforced expression of stemness-related factors in cancer cells may lead to contradictory results. Such factors, i.e., OCT4 and LIN28, are not only expressed in ESCs, adult stem cells, and iPSCs but are also highly expressed in cancer cells in the case of advanced-stage ovarian cancer [22]. c-MYC is a transcription factor that is constitutively and aberrantly expressed in over 70% of human cancers [23]. Therefore, C-MYC was shown to be oncogenic in a reprogramming process, and the activation of an MYC-dependent cancer enhancer was demonstrated to cause genetic mutations in human mammary epithelial cells [23]. Furthermore, SOX2 and the homeobox transcription factor NANOG were expressed in prostate, breast, lung, colon, and ovarian cancers [24,25]. KLF4 has also been reported to be oncogenic in osteosarcoma cells [26]. Hepatitis B virus X protein was reported to be associated with liver cancer stem cells (CSCs). This protein facilitated cell reprogramming by increasing the cellular levels of OCT4 and MYC [27]. KLF8 and O-linked N-acetylglucosamine transferase can promote CSC properties and cancer progression [28]. Human papillomavirus-16 E6 protein activated the expression of OCT4 and subsequently suppressed the transcription of p53 via a corepressor called nuclear receptor corepressor 1, which contributed to cervical cancer progression [29]. The novel developmental enhancer cluster Sox regulatory region SRR124–134 is highly accessible in most breast and lung tumors, where chromatin accessibility at these regions is correlated with *SOX2* overexpression and is regulated positively by forkhead box A1 protein and negatively by nuclear factor 1B [30]. Significantly increased expressions in SOX2, NANOG, and snail 1 and enhanced cell migration were found in claudin-6 transfected human adenocarcinoma cells [31]. These findings suggest that some stemness-related factors can be evaluated as proto-oncogenes or tumor markers.

In the present review, we discuss the current view of the risk of cancer formation due to reprogramming procedures in human normal and cancer cells. In addition, we summarize the potential strategies to suppress such risk of cancer initiation in iPSCs, iPCSCs, and organoids for their application in regenerative medicine and stem cell research.

## 2. Tools for Genetic Reprogramming

The methods for reprogramming somatic cells into iPSCs are summarized below. The main difference between these methods is the integration or non-integration into the host genomes.

## 3. Integration of Viral Vectors

Originally, pluripotency-related transcription factor genes, such as the combination of *OCT4*, *SOX2, KLF4*, and *c-MYC*, or *OCT4*, *SOX2*, *NANOG*, and *LIN28*, were introduced into somatic cells using retroviral or lentiviral vectors to generate iPSCs [1,2,7,8].

Random sites as multiple forms in the host DNA of the iPSCs, and it might disrupt MYC-dependent endogenous gene expression. Thus, the ideal methodology to generate iPSCs would involve reducing the number of pluripotency/stemness factors and using safe delivery viral systems without genome integration. To reduce this risk, some methods have been developed to improve the viral vector system using either the doxycycline-inducible expression system [32,33,34] or the Cre recombinase-locus of X-over P1 (Cre-loxP) system, which enables generating iPSCs without genome integration [35] or using various nonintegrating vectors [36].

## 4. Transposons

Two transposon-derived gene transfer systems, the piggyBac [37,38,39] and Sleeping Beauty transposon systems [40,41], have been generated as integration tools with a reduced risk of mutations. However, their reprogramming efficiency is lower than that of the viral vectors.

## 5. Nonintegrating Viral Vectors

The first nonintegrating protocol for genetic reprogramming was based on an adenovirus vector in 2008, wherein mouse hepatocytes were infected using a replication-incompetent adenovirus encoding OSKM [42]. Zhou and Freed also developed a method to generate iPSCs using human fibroblasts by introducing an adenovirus expressing OSKM [43]. However, using adenoviruses has the disadvantages of low infection efficiency and rapid removal of the transgenes from proliferative host cells.

The Sendai virus is an RNA virus containing single-stranded RNA as a negative sense that replicates in the cytoplasm without any DNA intermediary in the host cells. Because of these features, this virus is used worldwide for reprogramming various cell types, including fibroblasts [44], T-lymphocytes [45], and peripheral blood lymphoid cells. To ensure the removal of transgenes, temperature-sensitive vectors were used to allow the elimination of the virus at 37 °C [46], and a replication-deficient self-erasable Sendai virus recombinant vector with microRNA 302, which is expressed by pluripotent cells [47], was generated. Therefore, the Sendai virus vector is now commonly used as one of the standard methods for cell reprogramming, and it may even be transferrable to clinical settings.

## 6. Transfection of Linear DNA

Linear DNA was used for transfection using electroporation or a liposome system to reprogram cells using a polycistronic vector, which can express all inserted cDNAs from a single promoter. Although the simplicity of this method is attractive, it has low transfection efficiency. Yu et al. generated human iPSCs free of vector and transgene sequences using a single transfection vector with viral origin plasmid of origin oriP/Epstein–Barr nuclear antigen 1-based episomal vector [8]. Episomes are known as extrachromosomal DNAs that can replicate within cells autonomously. This system can be easily used to introduce genes encoding the reprogramming factors into host cells transiently. Polycistronic plasmids or those expressing two pairs of reprogramming factors, i.e., *OCT4/SOX2* and *KLF4/c-MYC*, can be used for transfection in somatic cells.

DNA minicircles are circular episomes containing a eukaryotic promoter and cDNAs of interest. The reprogramming of human adipose cells into iPSCs was reported by expressing OSKM in DNA minicircles [48,49]. The efficiency of reprogramming using this method is higher than that of using ordinal vectors. Although this method is attractive for its simplicity, it requires repeated transfection because it only permits a transient expression due to the progressive loss of DNA minicircles during each cell division.

## 7. Protein Delivery

Zhou et al. reported a successful cell reprogramming process mediated by recombinant proteins successfully into mouse fibroblasts [50]. A polyarginine (11R) protein domain linked to the C-terminus of each OSKM factor was introduced into cells. Then, valproic acid was added to generate iPSCs efficiently [51]. Human fibroblast cells were reprogrammed by using cell extracts generated by human embryonic kidney 293 cells expressing each of the four OSKM factors [52]. iPSCs were generated from somatic cells using reversible permeabilization drugs, such as streptolysin O [53] or cationic bola amphiphiles [54]. Although the main advantage of protein delivery is no integration into the host genome, this method is less attractive because of its lower reprogramming efficiency.

## 8. Chemical Molecules

Certain chemicals can not only enhance the reprogramming of somatic cells but also reprogram these somatic cells. For example, iPSCs were directly generated from mouse somatic cells using a cocktail of chemicals [4,55]. Mouse embryonic fibroblasts, neural progenitor cells, and small intestinal epithelial cells were simultaneously reprogrammed by only fine-tuning the concentrations of the chemical compounds. Moreover, bromodeoxyuridine can replace OCT4 as a reprogramming factor, and it was used along with several chemicals to produce iPSCs from mouse fibroblasts [56]. Although these small-molecule chemicals are used frequently to generate iPSCs for clinical use, their reprogramming efficiency is low.

## 9. RNA Delivery

The RNA delivery method has two advantages of using synthetic RNAs, namely, no integration into the genome, and their nuclear transfer is not required. Introducing the strategy of synthetic mRNA into host cells to produce pluripotency is the most footprint- and integration-free method. Moreover, the reprogramming efficiency of this method is highest compared with other nonviral, nonintegrating delivery systems [34]. Presently, RNA delivery appears to be one of the best strategies combining safety and efficiency, thereby making it a promising reprogramming strategy for future clinical applications.

## 10. Reprogramming Strategies Using Stemness-Related Genes

The signaling pathways underlying the differentiation of normal human stem cells into the ectoderm, mesoderm, mesendoderm, and definitive endoderm lineages are well understood [57]. Activation protein-1 (AP-1) family members regulated by p38/mitogen-activated protein kinase (MAPK) signaling are required to induce differentiation into the endoderm lineage while inhibiting cell fate shifting toward the mesoderm lineage. The c-Jun N–terminal kinase (JNK)–Jun signaling pathway reportedly blocked the exit from pluripotency via recruiting Jun to pluripotent enhancers along with OCT4, NANOG, and SMAD2/3 [58]. In addition, it has been reported that AP-1 enhancers were transiently bound during OSKM-induced cell reprogramming, which was linked to the disruption of the somatic transcription network [59]. Other transcription factors expressed in human pluripotent stem cells, such as nuclear respiratory factor 1, transcription factor AP-2, tumor suppressor protein TP53, and CCCTC-binding factor, appear to function critically in the early phase of mesoderm specification [60,61,62,63]. Chromatin immunoprecipitation sequencing analysis showed that the highest increase in H3K27Ac was observed on days 1 to 2 during the cell identity transition phase, resulting in the generation of novel super-enhancers after the first cell division. However, these super-enhancers were lost on day 3 and, thus, could not be detected during the later phase [64]. These epigenetic regulatory mechanisms are relevant for the developmental stage and for the homeostatic control of the stem cells under normal conditions and in diseases including cancer [60].

In reprogramming normal somatic cells, KLF4 promoted reprogramming via directly interacting with OCT4 and SOX2 [65], both of which function as transcriptional activators for reprogramming human fibroblasts [66]. In addition, SOX2 and KLF4 induced pluripotency without exogenous OCT4 in mouse embryonic fibroblasts (MEFs) and neural progenitor cells [67]. Furthermore, sirtuin 2 was essential for efficiently reprogramming MEFs to naïve states of pluripotency, in contrast to primed pluripotent states [68]. The inhibitors of mitogen-activated protein kinase 1/2 (MAPK1/2) and glycogen synthase kinase-3 and those of cell division protein kinases (cyclin-dependent kinase [CDK] 8/19) presented two approaches for establishing naïve pluripotency [69].

In addition, the homeobox protein NKX3-1, which is a downstream signaling protein in the interleukin-6/signal transducer and an activator of the transcription 3 network, is required for reprogramming and can replace OCT4 in mouse and human iPSC induction [70]. Huh et al. reported that suppressing the extracellular signal-regulated kinase (ERK)–serum responsive factor axis can facilitate iPSC formation in MEFs [71]. Li et al. found that the JNK–Jun pathway constituted a key barrier from pluripotency to definitive endoderm differentiation in human development. Moreover, JNK inhibitor treatment improved definitive endoderm and definitive endoderm-derived pancreatic and lung lineage differentiation and reduced the dose requirements for activin A [58].

Normoxia (21% O_2_) or hypoxia (5% O_2_) conditions were used to examine the effects of hypoxia on the production of iPSCs in MEFs and human dermal fibroblasts. Not only did the hypoxia-treated cells proliferate faster, but the generation of iPSCs increased significantly when cells were grown under 5% O_2_ during all induction periods [72]. A low oxygen tension (5% O_2_) condition resulted in the maintenance of highly proliferating ESCs and pluripotent human ESCs. Furthermore, the culture maintained at 20% O_2_ expressed significantly less OCT4, SOX2, and NANOG than at 5% O_2_ [73]. Hypoxia modulates the acetylation of histone by increasing H3 acetylation markers such as H3K27Ac and H3K9Ac. In addition, initiation of hypoxia and induction of hypoxia-inducible factor (HIF)-1α can increase the aggressiveness of neuroblastoma [74]. These techniques have been applied for reprogramming in normal cells but not in cancer cells.

## 11. Similar Characteristics between Cell Reprogramming and Cancer Initiation

Cancer-initiating cells arise from normal cells that have incurred genetic, epigenetic, and microenvironmental alterations [75]. The balance between cell proliferation/division and cell death is generally maintained in normal tissues and organs to preserve their integrity. Cells enter the initiation stage of cancer development on disruption of this balance. This hypothesis is generally accepted as the theory of tumorigenesis. Cancer initiation is also caused by epigenetic reprogramming that induces tumorigenic-enhancer reactivation in somatic cells and cancer cells [76,77,78]. p53 is a tumor suppressor protein that can regulate cell cycle arrest and apoptosis in response to DNA damage and prevent the generation of DNA mutation during cell division [79,80,81,82]. The pluripotency-related transcription factor c-MYC shows higher expression in various cancers, affects differentiation, and causes cancer development without expression of p53 [83,84]. During the process of iPSC establishment, cells acquire the abilities of dedifferentiation and indefinite proliferation. In addition, reprogramming enhancers might be involved in the shared function of the epigenetic change in reprogramming cancer-initiated chromatin markers during the iPSC production. Overlapping pathways may underlie these events of cell reprogramming and oncogenesis [85].

Chemical compounds and small molecules modifying the epigenome have been screened for their impact on cell reprogramming. The reprogramming efficiency was found to increase when using inhibitors of DNA methyltransferases like 5-azacytidine (5-AZA) [86], inhibitors of histone deacetylases like valproic acid [87] and sodium butyrate [88], and inhibitors of histone demethylases like parnate [89]. In fact, a combined treatment of parnate and the glycogen synthase kinase-3 inhibitor CHIR99021 allowed the generation of iPSCs from human keratinocytes using only two factors, such as OCT4 and KLF4 [89]. In addition, Huangfu et al. reported that valproic acid could replace the oncogene c-MYC or KLF4 to reprogram human primary fibroblasts using only two factors, OCT4 and SOX2 [87]. The combination of valproic acid and vitamin C, which induces DNA demethylation and alleviates cell senescence, increased the frequency of generation of iPSCs after reprogramming [90]. Sodium butyrate also enhanced the reprogramming efficiency of fetal and adult human fibroblasts [88]. Finally, late treatment with 5-AZA facilitated transiting cells to the pluripotent state, thus improving the reprogramming efficiency (Figure 1) [91].

When iPSCs are differentiated for therapeutic transplantation into patients, undifferentiated iPSCs may still exist in the differentiated cell mass, which might interfere with clinical application due to the risk of tumor formation. A previous study demonstrated that human iPSCs retained the methylation phase after the demethylation of CpG clusters in the promoter sequences of anti-oncogenes/oncogenes at the early stage of the disease. However, introducing pluripotency factors can repress the expression of oncogenes associated with the cancer phenotype and perturb epigenesis [92]. Pluripotency factors could be highly expressed in various types of cancers. Epigenetic differences between iPSCs and iPCSCs could affect their tumorigenicity. Therefore, the epigenetic memory of the iPCSCs might play a role in their tumorigenesis.

iPCSCs generated using glioblastoma-derived neural stem cells demonstrated a decrease in the ability to permeate into surrounding tissues. This finding suggested the repression of their aggressive features by resetting DNA methylation during the reprogramming procedure [93]. Therefore, whether pluripotency-related factors are promising targets for modulating the risk of carcinogenesis should be confirmed. Histone modifications, noncoding RNA alterations, and chromatin changes at the mutated sites of DNA are important traits that alter the oncogenic features [15].

A loss of function of cell cycle regulatory genes and continuous mutations in proto-oncogenes and/or tumor suppressor genes for initiating uncontrolled cell division are probably required for inducing cancerous transformation. Therefore, to reduce the risk of cancer initiation using the oncogenic gene-based pluripotency reprogramming protocols, new methodologies of cell reprogramming that can produce epigenetic and chromatin modifications will be needed.

## 12. Reducing Cancer-Initiation Risk in Established iPSCs and iPCSCs

The strategies for reducing the risk of tumorigenesis are described below in normal and CSCs, such as iPSCs and iPCSCs.

### 12.1. Selection of iPSC-Derived Differentiated Somatic Cells That Will Be Safe for Clinical Use

Experiments have been conducted to assess the quality of differentiated human NSCs derived from iPSCs after transplantation into immunodeficient animals to examine the possible risk of tumorigenesis in NSCs [94,95,96]. The introduction of an inducible caspase 9 (iCaspase 9) gene that resists tumorigenic transformation in human iPSCs was shown to prevent tumor formation after the transplantation of iPSC-derived somatic cells into mice with spinal cord injury [97,98]. iCaspase 9 treatment may be a strategy to inhibit the tumorigenic transformation of transplanted iPSC-derived cells in stem cell therapeutic transfers. However, this trial (AP1903) has not been reproduced [99] to create a system with a widely available immunosuppressive pharmaceutical to induce a suicide gene. Two constructs carry an inducible suicide gene, RapaCasp9-G or RapaCasp9-A, containing a single-nucleotide polymorphism (rs1052576) affecting the efficiency of endogenous caspase 9 [100]. These suicide genes are activated by rapamycin and are based on the fusion of human caspase 9 with a modified human FK-binding protein, allowing conditional dimerization. RapaCasp9-G- and RapaCasp9-A-expressing gene-modified T cells (GMTCs) were produced from healthy donors (HDs) and acute myeloid leukemia (AML) donors [101]. Another strategy may be using an inhibitor (YM 155) against the antiapoptotic factor survivin, which was shown to be effective in vitro and in adoptive transfers. In the study using YM 155, the survival of undifferentiated human iPSCs was inhibited without any toxicity for CD34^+^ cells. It was demonstrated that human iPSCs elucidated teratoma formation by YM 155 in immune-deficient mice after accomplishing cell transplantation studies [102]. Recently, the pathogenesis of pulmonary arterial hypertension was inhibited by YM155 was also reported [103]. Furthermore, the risk of cancer initiation of transplanted ESCs and iPSCs decreased when using etoposide, which is a DNA topoisomerase-II inhibitor, and purvalanol A, which is a CDK inhibitor, in cell transplantation trials [104,105]. Therefore, pretreatment of stem cells or iPSCs with apoptotic inducers and/or enzymes appears to help obtain safe clones for iPSC-based regenerative therapies (Figure 1).

### 12.2. Stemness Factors with Opposing Functions for Tumorigenicity during the Reprogramming of Cancer Cells to Maintain CSCs

Several methods have been applied to inhibit tumor formation during reprogramming in various types of cancers [106]. Miyoshi et al. [107] generated iPSC-like cells from gastrointestinal cancer cells using OSKM stemness factors, oncogenes, such as B cell lymphoma 2 and V-ki-ras2 Kirsten rat sarcoma 2 viral oncogene homolog (KRAS), and short-hairpin RNAs against tumor suppressor genes. The generated iPSC-like cells were sensitive to 5-fluorouracil and differentiation-inducing drugs and exhibited reduced tumor formation in immunodeficient mice. Furthermore, non-small cell lung cancer (NSCLC) cell lines were reprogrammed using OSKM to make iPCSCs that antagonized tumor-related genes in cancer cells both transcriptionally and epigenetically. This information about DNA methylation and the transcriptome might be possibly useful to reduce the tumor crisis [108].

Apart from the stemness factor-mediated repression of cancer, cancer-derived iPSC-like cells might be useful for personalized cancer treatments because these cells could provide a useful model for understanding the mechanism of drug resistance and discovery of a wide range of therapeutic agents targeting the genomic differences between individuals [79] (Figure 2).

As noted, several reports on reprogramming cancer cells using stemness factors demonstrated no change in tumorigenicity while exhibiting even stronger features of cancer initiation in derived iPSC-like cells [108,109,110,111,112]. Gastrointestinal cancer cells reprogrammed using OSKM were reported to show more aggressive cancer phenotypes than those of parental cells after long culturing [112]. It was suggested that derived iPSC-like cells could change the genetically unstable state through genetic or epigenetic alterations, including *c-MYC* activation (Figure 2).

C-Jun dimerization protein 2 (JDP2) is one of the AP-1/activating transcription factors. It controls various gene transcriptional activities and plays a crucial role in the transformation of normal cells into cancer cells [113,114]. JDP2, together with stemness factors, such as Jumonji-C domain-containing histone demethylase 1B, mitogen-activated protein kinase kinase 6, Gli-similar 1, NANOG, estrogen-related receptor β, and Sal-like transcription factor 4, was found to have the cell reprogramming potential to induce the transformation of somatic cells into chimera formation-competent iPSCs [113]. Furthermore, JDP2 could regulate stem cell characteristics in stem cells together with the Wnt signaling pathway [115,116,117]. We generated iPCSCs from a medulloblastoma cell line (DAOY cells) using OCT4 and JDP2 to induce cell reprogramming [109]. These iPSC-like cells displayed a stronger tumorigenic potential in immunodeficient mice than in the parental DAOY cancer cells. We observed the induction of a CSC-like state in reprogrammed cells via JDP2 overexpression [109]. The expression of JDP2 may promote cancer progression through Wnt signaling activation and induce CSC characteristics in cancer cells. We generated hepatoblastoma HepG2-derived iPCSCs in another study using OSKM and the knockdown vector shTP53 [111]. Such iPCSCs showed stemness properties and higher carcinogenesis after xenotransplantation compared with HepG2 cells. One cell line derived from the xenograft tumors exhibited aggressive phenotypes that increased cell growth, invasion, and chemoresistance. The expression of OCT4 and c-JUN was elevated in this cell line. Moreover, increased expression of OCT4 and c-JUN was confirmed in specimens from patients with liver cancer, and crosstalk between OCT4 and c-JUN was identified as underlying the tumorigenicity of this cancer. Therefore, these findings suggest that *JDP2* could be an oncogene in the medulla and liver. The exclusion of undifferentiated tumorigenic cells from a mass of stem cells or CSCs is essential for their application in therapy. However, the cell reprogramming strategy does not completely abolish the cancer-initiating risk of normal iPSCs. Even before the use of OSKM stemness factor overexpression for iPSC generation, one of the major factors, OCT4, had been recognized as a specific marker for detecting seminomas and embryonal carcinomas, which are germ cell tumors [118]. Thus, it is evident that stemness (reprogramming-inducing) factors play a definitive role in stem cell pluripotency and the interactions of various cofactors that can promote transcriptional adaptability in the cell development process, including tumorigenesis [119]. Many studies have reported the overexpression of reprogramming factors as markers for stemness in CSCs. c-MYC overexpression in immortalized mammary epithelial cells facilitated the onset of tumorigenesis via epigenetic reprogramming [23]. *KLF4* was found to be an oncogene in colon CSCs [120]. OCT4, SOX2, and NANOG were expressed in breast CSCs, and they played a role in maintaining these cells in the stem cell state [121,122]. Moreover, NANOG overexpression in breast CSCs stimulated the expression of other stemness factors, such as OCT4, KLF4, and *SOX2* [121]. In addition, hypoxia induced the expression of *NANOG* and HIF1α in breast CSCs by activating the telomerase reverse transcriptase gene. These findings indicate that simultaneous inhibition of reprogramming-inducing factors (genes), such as *OCT4*, *KLF4*, *SOX2*, *c-MYC*, and *NANOG*, as well as genes such as *JDP2*, telomerase reverse transcriptase gene, and *HIF1α*, leading to decreased stemness of residual CSCs, might be a promising strategy for cancer therapeutics [123,124].

As a different application of cancer cell reprogramming in cancer research, three-dimensional (3D) organoids produced by cancer-derived iPSC-like cells have been used to test anti-gastric cancer therapeutic agents [125,126,127,128,129]. Patient-derived organoids from cancer tissues are expected to accelerate the development of genome-wide research and novel personalized clinical therapies [129]. The application of organoids in this context is discussed below.

## 13. Additional Reprogramming Systems and Chemical Treatment with Small Molecules

Reprogramming of cells can be induced by chemical stimulation with small molecules that can promote the establishment of animal and human PSCs [4,11,12,13,14,15]. Chemical reprogramming has been recognized as a promising strategy for applying iPSCs in regenerative medicine [4].

We propose that reprogramming procedures that do not involve genetic materials can act as possible substitutes for generating safe iPSCs. Chemical exposure to certain small molecules can alter somatic cells’ fate to that of pluripotent stem cells [4,11,12,130,131]. As epigenetic regulators, small chemical molecules (AZA, valproic acid, and butyrate) have been used to enhance reprogramming efficiency and generate safe iPSCs without any genetic alterations [12,132].

Recently, human iPSCs were established using chemical epigenetic regulators (tranylcypromine, valproic acid, 3-deazaneplanocin A, EPZ004777, and UNC 0379) and cell signaling inhibitors (CHIR 99021, 616452, Y27632, and PD 0325901) to stimulate the pluripotency gene network [4]. It was demonstrated that inhibiting the JNK pathway, which is a major barrier to chemical reprogramming, facilitated the induction of pluripotency and self-renewal ability in somatic cells. Epigenetic silencing of the exogenous genes and enhanced chromatin remodeling during the maturation and stabilization phases of the cell reprogramming process were proposed to indicate the resetting of epigenetic alterations in reprogramming pluripotency-related genes [132]. It is unclear whether chemically reprogrammed human iPSCs have a reduced risk of cancer initiation after their regenerative transplantation into patients or animals. More studies are required to elucidate the molecular mechanisms and clarify the risk of tumorigenesis in cell reprogramming using chemical modulators.

In the murine iPSC-generating systems, excluding OCT4 from the OSKM factors allowed the generation of iPSCs with pluripotential competence equivalent to that of ESCs [133]. SKM-derived PSCs were confirmed to have the potential to generate all iPSC-derived mice, which was proven by the tetraploid embryo complementation test. The overexpression of OCT4 during cell reprogramming was hypothesized to be the cause of aberrant alterations in epigenesis, which leads to detrimental effects on the iPSCs. Removing OCT4 from the reprogramming factors might be preferable for applying iPSCs in regenerative medicine. However, the other three tumorigenic factors notably retain the risk of cancer initiation [5].

## 14. Modulating Factors for Increasing the Efficiency of Reprogramming

Reactive oxygen species (ROS), such as the superoxide anion (O^−^_2_)_,_ hydrogen peroxide (H_2_O_2_), and the hydroxyl radical (HO^−^), are produced by the partial reduction of oxygen. ROS are toxic by-products of aerobic metabolism that lead to cellular damage and cell death [134,135,136]. Higher levels of ROS reduce cell viability and decrease cell reprogramming efficiency. Conversely, ROS scavengers, such as vitamin C, decrease oxidative stress and increase cell reprogramming efficiency when added to the culture medium during mouse and human iPSC generation [89,137].

Vitamin C has anticancer properties due to its antioxidant and protective activities against oxidative stress-induced DNA damage [138]. Moreover, it leads to epigenomic remodeling in somatic cells by enhancing the function of Jumonji-C domain-containing histone demethylases and ten-eleven translocation proteins during cell reprogramming along with erasure of the epigenetic memory of the adult cell state [139].

In 2023, Japan’s Ministry of Health, Labor and Welfare announced that it would develop a new method for manufacturing industrial products. This methodology is expected to become a standard in the Organization for Economic Co-operation and Development’s International Evaluation Manual as an alternative to experiments on living animals. However, standardized, accurate data are needed to accomplish this task, for which it is a prerequisite to employ iPSCs generated using the same methodology worldwide.

## 15. Targeting of ROS-Induced Stem Cell Factors in Cancer

In this section, we briefly discuss the function of specific stemness factors, such as OCT4, SOX2, and NANOG, which are involved in ROS homeostasis, such as the aryl hydrocarbon receptor (AhR)–Nrf2 axis in cancer development. AhR affects the critical stages of tumorigenesis, such as initiation, progression, and metastasis [140], and anti-tumorigenesis, acting as a tumor suppressor [141,142]. There are significantly conflicting reports of AhR functioning as either a pro-tumorigenic or a tumor-suppressive factor in cancer. Bunaciu and Yen reported that retinoic acid-induced differentiation of leukemia cells correlated with increased AhR levels and decreased Oct4 levels, indicating a negative correlation between these two factors in CSCs [143]. Kynurenine derivatives, which are known as endogenous AhR ligands, can affect the transcription level of OCT4. Among them, Endogenous Trp derivatives 2-(10H-indole-30-carbonyl)-thiazole-4-carboxylic acid methyl ester (ITE) enhanced the binding of AhR to the OCT4 promoter and suppressed its transcription [144]. Alu retrotransposons located in the NANOG and OCT4 promoters containing AhR binding sites are transcribed by RNA polymerase III, and they repressed NANOG and OCT4 expression in differentiated carcinoma cells [145]. Liver carcinogenesis induced by diethyl nitrosamine produced strong carcinomas in all *AhR*^−/−^ mice but mostly premalignant adenomas in less than half of *AhR*^+/+^ mice. AhR-null tumoral tissues, but not their surrounding nontumoral parenchyma, had nuclear β-catenin and Axin2 overexpression. Nevertheless, OCT4 and NANOG were similarly expressed in *AhR*^+/+^ and *AhR*^−/−^ lesions. AhR was suggested to regulate liver repair and block tumorigenesis by modulating stem-like cells and β-catenin signaling [146]. NSCLC with *K-Ras^G12D^* mutations is one of the most prevalent types of lung cancer worldwide. Loss of AhR favors K-RasD12G-driven NSCLC. The pluripotency genes *Nanog*, *Sox2*, and *c-Myc* were also upregulated in *K-Ras^G12D^ AhR*^−/−^ lung tumors, and purified *K-Ras^G12D/+^ AhR*^−/−^ lung cells generated large numbers of organoids in culture that could subsequently differentiate into bronchioalveolar structures enriched in both pluripotency and stemness genes [147]. Moreover, upregulated SOX2 contributed to the stemness and metastasis of small cell lung cancer (SCLC) cells, while inhibition of the AhR signaling pathway blocked benzo(a)pyrene-induced protein kinase A (PKA) expression and downstream PKA/SOX2 axis. Our findings indicate benzo(a)pyrene exposure as a high-risk factor for SCLC and poor outcomes in patients, with the underlying mechanism being the activation of cancer stemness of SCLC cells via the AhR/PKA/SOX2 axis [148].

In general, the nuclear factor-erythrois-2-related factor 2 (Nrf2) has a contradictory role in cancers. Aberrant activation of Nrf2 is associated with poor prognosis. The constitutive activation of Nrf2 in various cancers induces prosurvival genes and promotes cancer cell proliferation through metabolic reprogramming, repression of cancer cell apoptosis, and enhancement of the self-renewal capacity of CSCs [149]. Nrf2, which is a master redox-sensitive transcription factor, plays an important role in the malignant transformation of lung carcinomas and the prevention of tumor initiation [150]. Cell migration is the first step for tumor progression, coinciding with cell invasion and metastasis [151]. Nrf2 activation was suggested to accelerate tumor metastasis [152]. DeNicola et al. [153] reported that the transcriptional start site of Nrf2 contained Jun and Myc binding sites. Therefore, the expression of Nrf2 and its downstream genes could be remarkably enhanced by activating the oncogenic alleles of *c-Myc*, *Braf*, and *Kras* (c-*Myc^ERT12^*, *Braf^V619E^*, and *Kras^G12D^*), leading to further reduction in the intracellular redox environment. Nrf2 promoter analysis in human NSCLC cells showed that a 2-O-tetradecanoylphorbol-13-acetate response element in the exon 1 regulatory region of Nrf2 was activated by Kras [154].

The major transcription factors OSKM also play roles in CSCs [155]. The expression of OCT4, which is a master regulator of cell pluripotency, is high in hepatocellular carcinoma and breast CSCs, and high OCT4 levels are associated with self-renewal, tumorigenicity, and chemoresistance of these CSCs [156,157]. Furthermore, OCT4 and cancerous inhibitors of protein phosphatase 2A are upregulated in testicular cancer cells and cell lines. The Kech-like ECH-associated protein 1(Keap1)–Nrf2–Heme oxygenase 1 (HO-1) signaling pathway was found to be regulated by OCT4 and a cancerous inhibitor of protein phosphatase 2A [158]. We also reprogrammed human medulloblastoma cells, such as DAOY, using JDP2 and OCT4. iPSC-like cells expressed stem cell-like characteristics, including alkaline phosphatase activity and expressions of some stem cell markers. However, such iPSC-like cells also proliferated rapidly, became neoplastic, and potentiated cell malignancy at a later stage in severe combined immunodeficiency mice [109].

Squamous tumors with overexpressing Nrf2 showed a molecular phenotype with SOX2/TP63 amplification, TP53 mutation, and CDKN2A loss. These immune cold NRF2 hyperactive diseases are associated with the upregulation of immunomodulatory NAMPT, WNT5A, SPP1, SLC7A11, SLC2A1, and PD-L1. Based on our functional genomics analyses, these genes represent candidate NRF2 targets, suggesting direct modulation of the tumor immune milieu [159].

NRF2, which is an oncogene and transcription factor, regulates gene expression to promote cancer progression, metabolic reprogramming, immune evasion, and chemoradiation resistance. Testing an optimized platform on 27 lung and upper aerodigestive cancer cell models revealed 35 NRF2-responsive proteins. In formalin-fixed paraffin-embedded head and neck squamous cell carcinomas, NRF2 signaling intensity positively correlated with NRF2-activating mutations and SOX2 expression [160].

These OSKM factors are related to the maintenance of ROS balance, which is mainly regulated by the expression of Nrf2 and AhR transcription factors. These two factors have two contradictory functions: one to stimulate cancer progression and the other to inhibit it. Thus, a strategy to regulate the expression of these factors is required to promote antioncogenesis and inhibit tumorigenesis. Therapeutically, making only these antioncogenesis signals from Nrf2 and AhR operate is important. The synergistic interaction between Nrf2 and AhR, called the “AhR–Nrf2 gene battery,” functions in detoxication to support cell survival [161]. The interaction of AhR with ketoconazole, which is an antifungal agent, induced a cytoprotective effect mediated by Nrf2 activation in cultured human keratinocytes. Accordingly, this gene battery is expected to have anticancer effects on the progression of malignant cells but not on normal cells. ROS production can be induced by AhR and Nrf2 activation. Cancer initiation can be induced by disrupting the balance of cellular ROS maintained by the AhR–Nrf2 gene battery [162]. We recently identified interactions of the AhR–Nrf2 battery via Jdp2 in normal MEFs and pancreatic cancer cell lines with mutations of Trp53 and Kras. Our findings suggest that Jdp2 acts as a tumor suppressor gene upstream of the AhR–Nrf2 gene battery in a spatiotemporal manner. Targeting AhR–Jdp2–Nrf2 interactions is a promising means of managing body homeostasis against internal and external stressors [163].

## 16. Discussion

This review highlights the current progress in reducing the tumorigenic risk of iPSC technologies, including reducing the tumorigenicity potential and promoting the death of abnormal cells. Possible strategies for reducing the tumorigenic risk are shown in Figure 3. The efforts outlined here are crucial for the clinical application of iPSC technology in cancer prevention. We have described the similarities and differences between carcinogenesis and reprogramming of cancer cells as well as normal cells. Although iPSC technology has facilitated the development of disease-specific models for distinct diseases, the recreation of carcinogenic conditions has been less successful. The efficiency of CSC reprogramming is rather low, and the genetic and epigenetic reprogramming needs to be clarified. The time periods and stages of cancer initiation and progression might be analyzed using their specific markers and epigenetic mechanisms.

Regarding CSCs and their niches, the following strategies were developed to eliminate the CSCs: (1) targeting CSCs using specific markers; (2) blocking the signal pathways of CSCs; (3) targeting quiescent CSCs; (4) using epigenetic therapy for depleting CSCs; (5) targeting CSC niches; and (6) using CSC-directed immunotherapy.

The SORE6 fluorescence reporter was constructed using six tandem repeats of the SOX2 and OCT4 response element from the proximal human NANOG promoter and has been used to detect prostate CSCs as a small fraction. Transcriptome and biochemical analyses identified phosphoinositide-3-kinase/AKT serine-threonine kinase (PI3K/AKT) signaling as critical for maintaining the SORE6^+^ population [165]. Other factors, such as IGF2 imprinting [166,167] and antivascular endothelial growth factor therapy by induction of the microseminoprotein, prostate-associated protein, are also critical in cancer development [168]. Using in vivo reprogramming techniques, the transient expression of reprogramming factors in Kras mutant mice was sufficient to generate the robust and persistent activation of ERK signaling in acinar cells and the development of precancerous lesions in Kras-mutated acinar cells [169]. The in vivo expression of OSKM provided primordial germ cell-associated features in adult somatic cells and promoted the development of human germ cell tumor-like cancers that exhibited trophoblast differentiation. The double sex- and mab-3-related transcription factor 1-mediated reprogramming played a crucial role in the propagation of germ cell-like tumor cells, which have the propensity to differentiate into trophoblasts [169].

Therefore, we propose targeting specific oncogenes or tumor suppressor genes, including the TP53 family, chromatin modifier CCCTC-binding factor, INK family of CDK inhibitors, RAS-G protein family, AP-1 family, TGF family, and their related signaling ERK/MAPK, p38/MAPK, and phosphoinositide-3-kinase/AKT signaling are critical for reprogramming cells to reduce the tumorigenic risk of iPSCs. The cascades of oxidative stress and its redox control machinery in the mitochondria and endoplasmic reticulum are also targets for cancer prevention. Why and how the abovementioned protein kinase family members regulate the tumorigenic risk during cell reprogramming remains to be addressed.

Thus, these trials to reduce the tumorigenic risk in the safe reprogramming of iPSCs for clinical application are very important and required for inhibiting future cancer risk in clinical use. Therefore, when applying this iPSC technology in clinical use, we must decrease the tumorigenic risk through our conceivable medical prevention.

## 17. Conclusions

This review addresses the current progress in reducing the tumorigenic risk of iPSC technologies. The tumorigenic potential of iPSCs can be decreased by optimizing the cocktail of reprogramming factors using the following strategies: (1) chemical reagent-induced reprogramming, (2) purification of iPSCs, (3) exogenous DNA-free vectors, and (4) nanoparticle and thermos-responsive polymers [170]. CSCs comprise a small subset of cancer cells with self-renewal, tumor initiation, and drug resistance capacities. The specific lineage markers of CSCs need to be clarified further, and the optimal culture conditions for their expansion need to be established. The environmental and immune cells within the CSC niches must be identified to isolate and analyze mutations. These issues can be addressed at the translational scale in clinical trials.

Recent progress in organoid research has generated high-throughput tools to study drug toxicities, disease-specific drug screenings, and drug development. Microfluidics has enabled the development of physiological tissue conditions to improve organoid cultures and might address organ-specific requirements in the future [171,172]. These organoids need to be characterized by deviating from iPSCs and assembling tissue models using different cell-sourced organoids to merging-on-a-chip technology. These advances are enabling the development of organ- and body-on-a-chip technology, which can be used to mitigate the ethical issues relating to animal use in research [129,173,174]. Recently, Deng’s group initiated new trials to decrease the time period and increase the efficiency of chemical reprogramming using human and mouse adipose-derived fibroblasts [4,175]. In addition, they generated islet cells from somatic cells using chemical reprogramming [175], and this strategy might be adapted to preclinical trials for reprogramming in the future. Such breakthrough strategies are required to reduce the tumorigenic potential of CSCs and cancer progenitor cells.

## Figures and Tables

**Figure 1 ijms-25-05177-f001:**
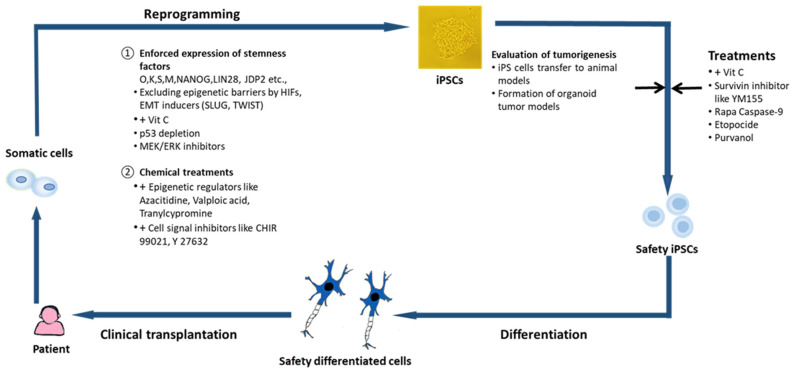
Methodologies of generating iPSCs and excluding tumorigenic iPSCs for therapeutic applications. Normal somatic cells are reprogrammed to iPSCs by the forced expression of stemness factors such as OCT4, KLF4, SOX2, c-MYC, NANOG, and LIN28 in cells. Supplementation of vitamin C, survivin inhibitor YM 155, CDK inhibitor etoposide, or purvalanol in iPSC cultures and RapaCasp9-G- and RapaCasp9-A-expressing gene-modified T cells were also generated. For the clinical transplantation safety of differentiated derivatives, tumor formation must be investigated using immunocompetent animals before clinical trials. Epigenetic regulators, such as AZA, valproic acid, and tranylcypromine, and cell signaling inhibitors, such as CHIR 99021 and Y27632, can be used as a substitute for genetic reprogramming factors to generate iPSCs. However, the tumorigenic risk of these chemical modulators in reprogramming requires further investigation.

**Figure 2 ijms-25-05177-f002:**
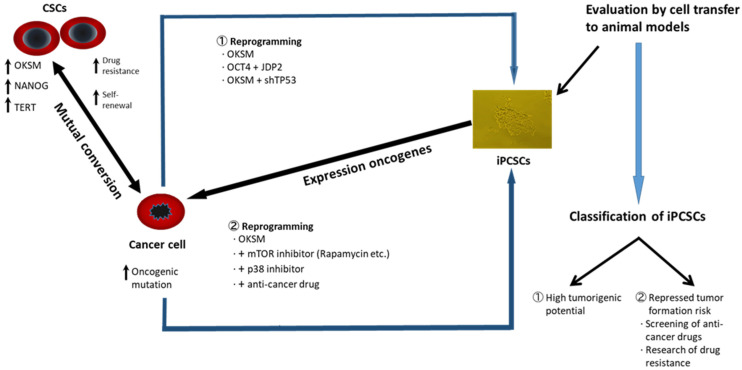
Cancer cell reprogramming procedures that generate iPCSCs with (1) high or (2) low tumorigenic potential. Tumorigenesis of iPCSCs is evaluated using cell transfer to animal models. Cancer cell reprogramming using forced expression of stemness factors and chemical molecules, such as anticancer drugs and cell signaling inhibitors, is useful to induce epigenetic alterations and change the tumorigenic state of the cancer cells. The degree of malignancy of iPCSCs in xenografts appears to depend on the cell of origin of the cancer. CSCs can maintain the quiescent state in cell division, which may be a signal for drug resistance to cancer treatments. The arrows indicate the cell fate conversion and the correlated features.

**Figure 3 ijms-25-05177-f003:**
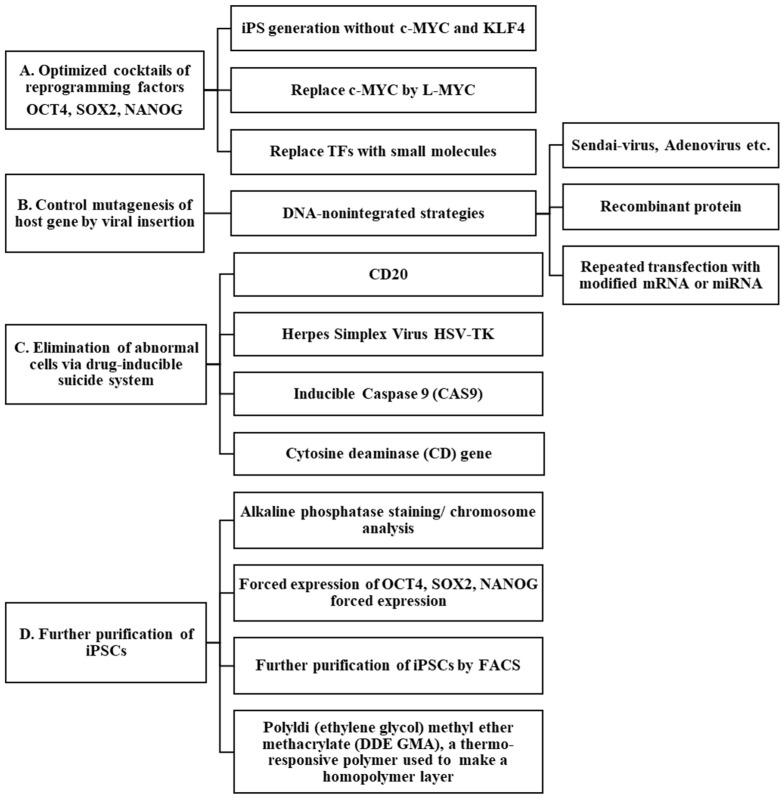
Possible strategies for reducing the tumorigenic risk in cell reprogramming. The methods to reduce tumorigenic risk are listed in five representative groups. Recently, chemical reprogramming has been described as the most powerful therapeutic invention method. This scheme is modified based on Zhong et al. [164], Copyright © The Author(s) 2022. Published by Oxford University Press on behalf of the West China School of Medicine & West China Hospital of Sichuan University.

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
