# Peer review of "Possible Strategies to Reduce the Tumorigenic Risk of Reprogrammed Normal and Cancer Cells"

_ijms, 2024, doi:10.3390/ijms25105177_

Round 1

Reviewer 1 Report

Comments and Suggestions for Authors

In the submitted manuscript Liu et al. review a number of different fields of research such as reprogramming to pluripotency, cancer stem cells, epigenetics, organoids, etc. Although all these topics are of interest none of them reviewed accurately. The title of the manuscript “Safe application of reprogramming strategies to avoid the tumorigenic risk in normal and cancer cell” is very intriguing itself because it is absolutely unclear how to avoid tumorogenicity of cancer cell which is tumorogenic by definition. Otherwise the cell is not cancerous. In the abstract Authors denominate the subject of their review as the recent progress in research focused on decreasing the tumorigenic risk of induced pluripotent stem cells (iPSCs) or iPSC-derived organoids for therapeutic application. iPSCs are indeed very promising source of autologous or allogenic somatic cells (retinal cells, dophamergic neurons, and some others) for transplantation and a number of clinical trials are ongoing now. However, pluripotent stem cells (iPSCs, ESCs) themselves may cause tumor-like structures especially in immunocompromised organisms thus direct therapeutic use of undifferentiated PSCs is rather difficult to suppose presently.  Actually PSCs transplantation into immunedeficient mice is a routine and indispensable test for the pluripotency confirmation.  Any manipulations with primary cells in vitro and reprogramming in particular may introduce some mutations that will lead to cell transformation. Thus it is important for each therapeutically significant iPSC line to validate not only stemness genes integrity but also genome stability after reprogramming and further differentiation into particular therapeutic cell (or organoid) product. Overall the topic of the manuscript is very important however the paper is overloaded by the information unrelated to the defined subject of the review. Sections “Removal of epigenetic barriers in cancer stem cells” (nothing about reprogramming),  “Identification and lineage tracing of CSCs to reduce the cancer-initiation risk”,  “ Cancer risk prevention studies using cancer organoids” “Targeting of ROS-induced stem cell factors in cancer” are out of the scope of the safe application of reprogramming strategies.  Section “Reprogramming strategies using stemness-related genes” although has some relation to reprogramming however considering that practically none of the cited genes are in use now should be shortened.  There are also some unclear points:

1.Line 61-63. There is nothing in the ref 13 about p53 and frequency of tumorogenesis

2. Line 73 Fig1 legend- For clinical transplantation safety of differentiated derivatives must be studied  as it currently performed in clinical trials. While any iPSC line must form tumor-like structures into immunocompromized animals.

3.Line 83-84. 50% of cancers are due to p53 mutations therefore reprogramming should be not difficult but enhanced

4.Line 91-94. Unclear. Rephrase.

5. Line 95. c-Myc is a transcription factor that is constitutively and aberrantly expressed in over 70% of human cancers not only during reprogramming and mammary epithelial cells.

6. Line 130. How does inducible system that uses viral vectors improve genome integration?

7. Line181-189.  From this section Authors conclude that mouse iPSCs could be used in clinics. Explain.

8.  Line 249-253. Cited review ref 74  is about tumor metabolism and is rather unrelated to the theory of tumorogenesis.

Line 254. “Cancer initiation is also caused by the epigenetic reprogramming that induces tumor….” “Also”- wthat’s else is involved? Why reprogramming? Is cancer a developmental program? As far as I know genetic alterations are the primary steps of cell transformation.

Line 279. Most indicated chemicals make chromatin more accessible for transcription factors to initiate reprogramming but at the same time DNA is also getting more accessible for damage. I dare to say but even removal might be too late. Uneven connection between timing and mutations.

Line 281-284. It looks that Authors suggest to transplant to patients cells derived from reprogrammed human sarcoma. What is the purpose of such transplantation and comparison with somatic cells differentiated from iPSCs? Overall it is unclear on what basis Authors try to draw parallels between genetically transformed cancer cells with modified epigenetics and reprogrammed to pluripotency normal somatic cells?

Line 425  Selection of safe iPSC-derived clones.  Not quite clear from the heading what clones Authors mean, however it is of importance to select iPSC-derived differentiated somatic cells that will be safe for clinical usage. Surprisingly that Authors did not analyze such technologies and their outcome.

It is unclear how the section “Stemness factors with opposing functions for tumorigenicity during the reprogramming of cancer cells to maintain CSCs” is connected with the decreasing the tumorigenic risk of induced pluripotent stem cells (iPSCs) or iPSC-derived organoids for therapeutic application, it is not clear from the section and it contains a lot of incorrect statement.

Line 451Furthermore, non-small cell lung cancer (NSCLC) cell 451 lines were reprogrammed using OSKM to make iPCSCs that antagonized tumor-related 452 genes in cancer cells both transcriptionally and epigenetically, resulting in declined tu-453 morgenicity in the reprogrammed cells [137]. “ Tumorogenicity was not studied in this paper.

Line 454” In similar experiments on human sarcoma 454 cell lines, stemness factors could repress the expression of cancer phenotypes, modify the 455 epigenetic landscape, and alter the expression of cancer-related genes [138].” Unrelated reference.

Discussion and conclusion sections.  Fig 3 is absolutely unclear. Authors state that chemical reprogramming is a very powerful method, however I would recommend to exclude it completely from the text of the manuscript or change the aim of the paper – iPSC and iPSC derived cell therapeutic application- because I doubt that mouse cells will be clinically applied.  

Comments on the Quality of English Language

In the submitted manuscript Liu et al. review a number of different fields of research such as reprogramming to pluripotency, cancer stem cells, epigenetics, organoids, etc. Although all these topics are of interest none of them reviewed accurately. The title of the manuscript “Safe application of reprogramming strategies to avoid the tumorigenic risk in normal and cancer cell” is very intriguing itself because it is absolutely unclear how to avoid tumorogenicity of cancer cell which is tumorogenic by definition. Otherwise the cell is not cancerous. In the abstract Authors denominate the subject of their review as the recent progress in research focused on decreasing the tumorigenic risk of induced pluripotent stem cells (iPSCs) or iPSC-derived organoids for therapeutic application. iPSCs are indeed very promising source of autologous or allogenic somatic cells (retinal cells, dophamergic neurons, and some others) for transplantation and a number of clinical trials are ongoing now. However, pluripotent stem cells (iPSCs, ESCs) themselves may cause tumor-like structures especially in immunocompromised organisms thus direct therapeutic use of undifferentiated PSCs is rather difficult to suppose presently.  Actually PSCs transplantation into immunedeficient mice is a routine and indispensable test for the pluripotency confirmation.  Any manipulations with primary cells in vitro and reprogramming in particular may introduce some mutations that will lead to cell transformation. Thus it is important for each therapeutically significant iPSC line to validate not only stemness genes integrity but also genome stability after reprogramming and further differentiation into particular therapeutic cell (or organoid) product. Overall the topic of the manuscript is very important however the paper is overloaded by the information unrelated to the defined subject of the review. Sections “Removal of epigenetic barriers in cancer stem cells” (nothing about reprogramming),  “Identification and lineage tracing of CSCs to reduce the cancer-initiation risk”,  “ Cancer risk prevention studies using cancer organoids” “Targeting of ROS-induced stem cell factors in cancer” are out of the scope of the safe application of reprogramming strategies.  Section “Reprogramming strategies using stemness-related genes” although has some relation to reprogramming however considering that practically none of the cited genes are in use now should be shortened.  There are also some unclear points:

1.Line 61-63. There is nothing in the ref 13 about p53 and frequency of tumorogenesis

2. Line 73 Fig1 legend- For clinical transplantation safety of differentiated derivatives must be studied  as it currently performed in clinical trials. While any iPSC line must form tumor-like structures into immunocompromized animals.

3.Line 83-84. 50% of cancers are due to p53 mutations therefore reprogramming should be not difficult but enhanced

4.Line 91-94. Unclear. Rephrase.

5. Line 95. c-Myc is a transcription factor that is constitutively and aberrantly expressed in over 70% of human cancers not only during reprogramming and mammary epithelial cells.

6. Line 130. How does inducible system that uses viral vectors improve genome integration?

7. Line181-189.  From this section Authors conclude that mouse iPSCs could be used in clinics. Explain.

8.  Line 249-253. Cited review ref 74  is about tumor metabolism and is rather unrelated to the theory of tumorogenesis.

Line 254. “Cancer initiation is also caused by the epigenetic reprogramming that induces tumor….” “Also”- wthat’s else is involved? Why reprogramming? Is cancer a developmental program? As far as I know genetic alterations are the primary steps of cell transformation.

Line 279. Most indicated chemicals make chromatin more accessible for transcription factors to initiate reprogramming but at the same time DNA is also getting more accessible for damage. I dare to say but even removal might be too late. Uneven connection between timing and mutations.

Line 281-284. It looks that Authors suggest to transplant to patients cells derived from reprogrammed human sarcoma. What is the purpose of such transplantation and comparison with somatic cells differentiated from iPSCs? Overall it is unclear on what basis Authors try to draw parallels between genetically transformed cancer cells with modified epigenetics and reprogrammed to pluripotency normal somatic cells?

Line 425  Selection of safe iPSC-derived clones.  Not quite clear from the heading what clones Authors mean, however it is of importance to select iPSC-derived differentiated somatic cells that will be safe for clinical usage. Surprisingly that Authors did not analyze such technologies and their outcome.

It is unclear how the section “Stemness factors with opposing functions for tumorigenicity during the reprogramming of cancer cells to maintain CSCs” is connected with the decreasing the tumorigenic risk of induced pluripotent stem cells (iPSCs) or iPSC-derived organoids for therapeutic application, it is not clear from the section and it contains a lot of incorrect statement.

Line 451Furthermore, non-small cell lung cancer (NSCLC) cell 451 lines were reprogrammed using OSKM to make iPCSCs that antagonized tumor-related 452 genes in cancer cells both transcriptionally and epigenetically, resulting in declined tu-453 morgenicity in the reprogrammed cells [137]. “ Tumorogenicity was not studied in this paper.

Line 454” In similar experiments on human sarcoma 454 cell lines, stemness factors could repress the expression of cancer phenotypes, modify the 455 epigenetic landscape, and alter the expression of cancer-related genes [138].” Unrelated reference.

Discussion and conclusion sections.  Fig 3 is absolutely unclear. Authors state that chemical reprogramming is a very powerful method, however I would recommend to exclude it completely from the text of the manuscript or change the aim of the paper – iPSC and iPSC derived cell therapeutic application because I doubt that mouse cells will be clinically applied.  

Author Response

[Reviewer 1]

In the submitted manuscript Liu Lin et al. review a number of different fields of research such as reprogramming to pluripotency, cancer stem cells, epigenetics, organoids, etc. Although all these topics are of interest none of them reviewed accurately. The title of the manuscript “Safe application of reprogramming strategies to avoid the tumorigenic risk in normal and cancer cell” is very intriguing itself because it is absolutely unclear how to avoid tumorogenicity of cancer cell which is tumorogenic by definition. Otherwise, the cell is not cancerous. In the abstract Authors denominate the subject of their review as the recent progress in research focused on decreasing the tumorigenic risk of induced pluripotent stem cells (iPSCs) or iPSC-derived organoids for therapeutic application. iPSCs are indeed very promising source of autologous or allogenic somatic cells (retinal cells, dophamergic neurons, and some others) for transplantation and a number of clinical trials are ongoing now. However, pluripotent stem cells (iPSCs, ESCs) themselves may cause tumor-like structures especially in immunocompromised organisms thus direct therapeutic use of undifferentiated PSCs is rather difficult to suppose presently.  Actually, PSCs transplantation into immunedeficient mice is a routine and indispensable test for the pluripotency confirmation.  Any manipulations with primary cells in vitro and reprogramming in particular may introduce some mutations that will lead to cell transformation. Thus, it is important for each therapeutically significant iPSC line to validate not only stemness genes integrity but also genome stability after reprogramming and further differentiation into particular therapeutic cell (or organoid) product.

Overall, the topic of the manuscript is very important however the paper is overloaded by the information unrelated to the defined subject of the review. Sections “Removal of epigenetic barriers in cancer stem cells” (nothing about reprogramming), “Identification and lineage tracing of CSCs to reduce the cancer-initiation risk”, “Cancer risk prevention studies using cancer organoids” “Targeting of ROS-induced stem cell factors in cancer” are out of the scope of the safe application of reprogramming strategies.  Section “Reprogramming strategies using stemness-related genes” although has some relation to reprogramming however considering that practically none of the cited genes are in use now should be shortened.  There are also some unclear points:

Response: As suggested by the reviewer, I revised the text to accommodate most of the criticisms and comments including the deletion of some agenda. I also corrected my typing errors in the text.

  1. Line 61-63 (New line 70-79). There is nothing in the ref 13 about p53 and frequency of tumorogenesis

Response: As suggested by the reviewer, I revised the reference and the corresponding text as follows.

In some cases, abnormal or deleted p53 significantly increased not only the reprogramming efficiency of iPSC-like cells (referred to as induced pluripotent cancer stem cells, iPCSCs) but also the tumorigenicity of iPCSCs, as observed in mice derived from iPSCs with p53 knockout [13,14]. Furthermore, suppression of the expression of anti-oncogenes was observed to not only enhance the reprogramming efficiency of cells but also adversely increase the risk of tumorigenesis after transplantation [15] (Fig. 1). These findings, together with many other similar findings, strongly suggest that tumor reprogramming, and iPSC generation share similar pathways. Therefore, the risk of tumorigenesis in iPSC-based stem cell applications is a major concern. We speculate that cancer stem cells may be generated through reprogramming. 

  1. Line 73 (New line 82) Fig1 legend- For clinical transplantation safety of differentiated derivatives must be studied as it currently performed in clinical trials. While any iPSC line must form tumor-like structures into immunocompromized animals.

Response: I deleted this sentence and added the revised sentence to the text (Figure legend in Figure 1) as follows (New Line 86-87).

For the clinical transplantation safety of differentiated derivatives, tumor formation must be investigated using immunocompetent animals before clinical trials.

  1. Line 83-84 (Ne line 96-98). 50% of cancers are due to p53 mutations therefore reprogramming should be not difficult but enhanced.

Response: I apologize that I am unable to understand the meaning of this comment. However, I revised lines 96–98 as follows.

The reprogramming efficiency is lower in cancer cells than in normal cells because the cancer epigenome and chromosomal changes or gene mutations are present in cancer cells.

  1. Line 91-94 (Line 104-107). Unclear. Rephrase.

Response: I rephrased this section as follows.

Moreover, the enforced expression of stemness-related factors in cancer cells may lead to contradictory results. Such factors (i.e., OCT4 and LIN28) are not only expressed in ESCs, adult stem cells, and iPSCs but are also highly expressed in cancer cells in the case of ovarian cancer with advanced stage [22].

  1. Line 95 (line 107-111).

Response: I added the following sentence in the text:

c-MYC is a transcription factor that is constitutively and aberrantly expressed in over 70% of human cancers [23].

  1. Line 130 (line 142-146). How does inducible system that uses viral vectors improve genome integration?

Response: I revised the sentence in line 130. The system using viral vectors in references 31–33 (new references 33–36) is described below. These studies used nonintegrating viral reprogramming vectors.

  1. Wernig, M.; Lengner, C.J.; Hanna, J.; Lodato, M.A.; Steine, E.; Foreman, R.; Staerk, J.; Markoulaki, S.; Jaenisch, R. A drug-inducible transgenic system for direct reprogramming of multiple somatic cell types. Nat Biotechnol 2008, 26, 916–924.
  2. Maherali, N.; Ahfeldt, T.; Rigamonti, A.; Utikal, J.; Cowan, C.; Hochedlinger, K. A high-efficiency system for the generation and study of human induced pluripotent stem cells. Cell Stem Cell 2008, 3, 340–345.
  3. Hockemeyer, D.; Soldner, F.; Cook, E.G.; Gao, Q.; Mitalipova, M.; Jaenisch, R. A drug-inducible system for direct reprogramming of human somatic cells to pluripotency. Cell Stem Cell 2008, 3, 346–353.
  4. Chang, C.W.; Lai, Y.S.; Pawlik, K.M.; Liu, K.; Sun, C.W.; Li, C.; Schoeb, T.R.; Townes, T.M. Polycistronic lentiviral vector for “hit and run” reprogramming of adult skin fibroblasts to induced pluripotent stem cells. Stem Cells 2009, 27, 1042–1049.
  5. Bailly, A.; Milhavet, O.; Lemaitre, J.M. RNA-based strategies for cell reprogramming toward pluripotency. Pharmaceutics 2022, 14, 317.

Reference 33 described the generation of genetically homogeneous “secondary” somatic cells carrying deoxycycline (dox)-inducible reprogramming factors OSKM or NANOG.  These cells were produced by infecting fibroblasts with dox-inducible lentiviruses, reprogramming by dox addition, selecting induced pluripotent stem cells, and producing chimeric mice. Cells derived from these chimeras undergo reprogramming on dox exposure without the need for viral infection with efficiencies 25- to 50-fold greater than those observed using direct infection and drug selection for pluripotency marker reactivation. Similarly, references 34-35 used a reverse tetracycline transactivator (rtTA) driven by a ubiquitin promoter cloned into a lentiviral vector. To generate human iPSCs, neonatal foreskin fibroblasts and keratinocytes with lentiviruses containing the rtTA and either four (i.e., OCT4, SOX2, cMYC, and KLF4 for fibroblasts) or five reprogramming factors (the abovementioned four factors plus NANOG for both fibroblasts and keratinocytes) were used. In this study, dox-inducible lentiviral vectors carrying cDNAs encoding the reprogramming factors were used. The cells were simultaneously infected with a constitutively active lentivirus expressing the reverse tetracycline transactivator (FUW-M2rtTA). The infected cells were cultured in the presence of dox and iPSCs. Human ES cell-like morphology was detected after approximately 1 and 2 months in four- and three-factor fibroblast cultures, respectively. In reference 36, the so-called “hit and run” vector using the insertion sites that remnant 291-bp LTRs containing a single loxP site did not interrupt coding sequences, promoters, or known regulatory elements. After deletion of the vector, small remnant fragments (291 bp) remained in the iPSC genome. However, these DNA fragments did not contain promoter or enhancer sequences and did not interrupt coding sequences, promoters, or regulatory elements. Thus, the probability of insertional activation or inactivation of endogenous genes was extremely low. The final vectors were the RNA vectors as usual. However, these studies did not focus on viral genome integration, and strategies for safe reprogramming are required (ref 37).

  1. Line181-189 (line 185-194).  From this section Authors conclude that mouse iPSCs could be used in clinics. Explain.

Response: I apologize for the misunderstanding. I did not mean that mouse iPSCs could be used in clinics. References 50 and 51 are related to mouse iPSCs; however, reference 52 is related to human iPSCs.

The mouse studies are basic translational studies that could be used in the future for human iPSC replacements. For translational research, human cells or animal models using their own iPSCs are used. We must set up the human cells and then design the translational research using human iPSCs in animals, organoids, and finally patients. Thus, even though I mentioned two references for mouse iPSCs, this basic idea is useful for human iPSC strategies.

  1. Line 249-253 (new Line 264-270). Cited review ref 74 (ref 76) is about tumor metabolism and is rather unrelated to the theory of tumorogenesis.

Response: I agree with your comment. Thus, I replaced reference 74 (new reference 76) with a new reference to describe the balance of genetic and epigenetic alterations as follows.

  1. Takejima, H., and Ushijima, T. Accumulation of genetic and epigenetic alterations in normal cells and cancer risk. npJ Precis Oncol 2019, 3, 7.

Line 254 (new Line 268-270). “Cancer initiation is also caused by the epigenetic reprogramming that induces tumor….” “Also”- what’s else is involved? Why reprogramming? Is cancer a developmental program? As far as I know genetic alterations are the primary steps of cell transformation.

Response: I understand your criticism; however, I feel that it is not justified because the statement “cancer initiation is caused by epigenetic reprogramming.” is validated by other observation including the reference by Vicente-Duenas, C. et al., Trends Cancer 2018, 4, 408–417. Moreover, epigenetic reprogramming has been observed to induce tumorigenic reactivation in somatic and cancer cells (Abatti, L.E. et al., Nucleic Acids Res 2023, 51, 10109–10134; Poli, V. et al., Nat Communication 2018, 9, 1024).

Epigenetic alterations are also required beforehand for genetic alterations to occur. Epigenetic and genetic alterations coupled with each other, and their alterations might cause the initiation of cancer transformation. We also cited the abovementioned reference by Takejima and Ushijima (ref 76). Thus, the reference should be 77–79. In addition, the above three references also showed this correlation.

Epigenetic alterations and chromatin changes play roles in the alteration of genomes. Epigenetic reprogramming using stemness genes or other inducers might be the primary event for such genomic alteration. Then, the driver mutations in genes, such as p53, Ras, APC, BRACA, and EGFR, and DNA mutations are the initiation events of cell transformation, as you pointed out. Thus, epigenetic reprogramming not only involves reprogramming of the epigenome but also that of the genome.

 My response regarding the questions “what else is involved? Why reprogramming? Is cancer a developmental program?” is following.

For example, the reprogramming factors used to change the epigenome via histone methylations, such as H3K9me3 to H3K4me3 and H3K27me3 to H3K27Ac, or DNA methylations around GC clusters are involved. Thus, we think that the cancer initiation was triggered by these epigenetic reprogramming, and then the DNA genetic mutation might occur as the results of driver mutations. Thus, the normal cells or the cancer cells might occur to the cancer stem cells at the initiation stage. The surrounding microenvironmental cells around cancer stem cells might change to the mutated cells to alter the cancer stem cells into the more aggressive cancer stem cells.  Thus, these epigenome changes and genetic changes in not only the cancer stem cells but also the stem cell niches microenvironment. This event might happen at the initial stage of the cancer triggering. We cannot say that the caner is a developmental program. However, we believe that the epigenetic reprogramming is the first triggering to alter the transcriptional alteration as the cancer initiation, which results in the genetic mutation during the cancer triggering.

Line 279. Most indicated chemicals make chromatin more accessible for transcription factors to initiate reprogramming but at the same time DNA is also getting more accessible for damage. I dare to say but even removal might be too late. Uneven connection between timing and mutations.

Response: I agree with this comment because the mutation and removal time are not even related to each other. Thus, I deleted this sentence.

Line 281-284 (New Line 295 -304). It looks that Authors suggest to transplant to patients cells derived from reprogrammed human sarcoma. What is the purpose of such transplantation and comparison with somatic cells differentiated from iPSCs? Overall it is unclear on what basis Authors try to draw parallels between genetically transformed cancer cells with modified epigenetics and reprogrammed to pluripotency normal somatic cells?

Line 285-292 (New Line 295-304). Authors try to draw parallels between genetically transformed cancer cells with modified epigenetics and reprogrammed to pluripotency normal somatic cells?

Response: I apologize for the misunderstanding. The purpose of this paragraph was to describe the transplantation of iPSCs into a patient’s tissues derived from reprogrammed human sarcoma. I did not intend to draw parallels between cancer cells with epigenetic modifications and reprogrammed normal somatic cells during transplantation.

I attempted to focus on the heterogeneity of the iPCSCs that were reprogrammed. I revised this paragraph as follows.

When iPSCs are differentiated for therapeutic transplantation into patients, undifferentiated iPSCs may still exist in the differentiated cell mass, which might interfere with clinical application due to the risk of tumor formation. A previous study demonstrated that human iPSCs retained the methylation phase after the demethylation of CpG clusters in the promoter sequences of anti-oncogenes/oncogenes at the early stage of the disease, However, introducing pluripotency factors can repress the expression of oncogenes associated with the cancer phenotype and perturb epigenesis [93]. Pluripotency factors could be highly expressed in various types of cancers. Epigenetic differences between iPSCs and iPCSCs could affect their tumorigenicity. Therefore, the epigenetic memory of the iPCSCs might play a role in their tumorigenesis.

Line 325 (New Line 322-349).  Selection of safe iPSC-derived clones.  Not quite clear from the heading what clones Authors mean, however it is of importance to select iPSC-derived differentiated somatic cells that will be safe for clinical usage. Surprisingly that Authors did not analyze such technologies and their outcome.

Response: As suggested by the reviewer, I changed the title to a new one. The reviewer commented that I did not analyze these technologies and the validity of iPSC-derived clones. Thus, I revised the text to incorporate a recent study instead of the iCaspase 9 trial [98, 99], which was reported to have failed. I think this technology of iCaspase 9 [named AP1903, (the fusion of human caspase 9 to a modified human FK-binding protein, allowing conditional dimerization using small molecule drug)] was unsuccessful for the validation for the trial of clinical application [100]. Unfortunately, AP1903 is not a licensed pharmaceutical agent and cannot be used for therapy. Thus, to create a system with a widely available immunosuppressive pharmaceutical to induce a suicide gene, two constructs carrying an inducible suicide gene, RapaCasp9-G or RapaCasp9-A, containing a single-nucleotide polymorphism (rs1052576) affecting the efficiency of endogenous caspase 9 were used [new reference 101]. These suicide genes are activated by rapamycin and based on the fusion of human caspase 9 with a modified human FK-binding protein, allowing conditional dimerization. RapaCasp9-G- and RapaCasp9-A-expressing gene-modified T cells (GMTCs) were produced from healthy donors (HDs) and acute myeloid leukemia (AML) donors [new reference 102].

The references were 101 to 105.

  1. Straathof, K.C.; Pulè, M.A.; Yotnda, P.; Dotti, G.; Vanin, E.F.; Brenner, M.K.; Heslop, H.E.; Spencer, D.M.; Rooney, C.M. An inducible caspase 9 safety switch for T-cell therapy. Blood 2005, 105, 4247–4254.
  2. Itakura, G.; Kawabata, S.; Ando, M.; Nishiyama, Y.; Sugai, K.; Ozaki, M.; Iida, T.; Ookubo, T.; Kojima, K.; Kashiwagi, R.; et al. Fail-Safe System against potential tumorigenicity after transplantation of iPSC derivatives. Stem Cell Reports 2017, 8, 673–684.
  3. Garget, T.; Brown, M.P. The inducible caspase-9 suicide system as a “safety switch” to limit in-target, off-tumor toxicities of chimeric antigen receptor T cells. Front Pharmacol 2014, 5, 254.
  4. Stavrou, M.; Philip, B.; Traynor-White, C.; Davis, C.G.; Onuoha, S.; Cordoba, S.; et al. A rapamycin-activated caspase 9-based suicide gene. Mol Therapy 2018, 26, 1266–1276.
  5. Bouquet, L.; Bôle-Richard, E.; Warda, W.; Neto Da Rocha, M.; Trad, T.; Nicod, C.; Haderbache, R.; Genin, D.; Ferrand, C.; Deschamps, M. RapaCaspase-9-based suicide gene applied to the safety of IL-1RAP CAR-T cells. Gene Therapy, 2023, 30, 706–

It is unclear how the section “Stemness factors with opposing functions for tumorigenicity during the reprogramming of cancer cells to maintain CSCs” is connected with the decreasing the tumorigenic risk of induced pluripotent stem cells (iPSCs) or iPSC-derived organoids for therapeutic application, it is not clear from the section, and it contains a lot of incorrect statement.

Response: The reviewer commented that this section contained a lot of incorrect statements and the connection between this section and decreasing the tumorigenic risk of iPSCs or iPSC-derived organoids for therapeutic application is unclear. However, the reviewer may have extrapolated my statement to direct clinical application. I did not state that these trials can lead to direct application because more basic and translational studies, with many basic experiments, are needed to show the validity of this technology using the stemness factors in clinical application. Thus, I think that this section is useful for the future application of this technology to clinical translation and interesting for the readers of the journal. (New Line 322-349)

Line 451 (New Line 360-364) “Furthermore, non-small cell lung cancer (NSCLC) cell 451 lines were reprogrammed using OSKM to make iPCSCs that antagonized tumor-related 452 genes in cancer cells both transcriptionally and epigenetically, resulting in declined tu-453 tumorgenicity in the reprogrammed cells [137]. “Tumorogenicity was not studied in this paper.

Response: I agree with the reviewer’s comment. These candidate genes are useful for future applications. Thus, I changed this section [New Line 360 to 363] to “This information about the DNA methylation and transcriptome might be possibly useful to produce the new strategy to reduce the tumorigenesis”.

Furthermore, non-small cell lung cancer (NSCLC) cell lines were reprogrammed using OSKM to make iPCSCs that antagonized tumor-related genes in cancer cells both transcriptionally and epigenetically. This information about the DNA methylation and the transcriptome might be possibly useful to produce the new strategy to reduce the tumorigenesis [109].

Line 454 (Line 356)” In similar experiments on human sarcoma cell lines, stemness factors could repress the expression of cancer phenotypes, modify the epigenetic landscape, and alter the expression of cancer-related genes [138].” Unrelated reference.

Response: As suggested by the reviewer, I deleted this reference and this sentence.

In similar experiments on human sarcoma cell lines, stemness features could repress the expression of cancer penotypes. Modify the epigenetic landscape. And alter the expressions of cancer related gene [138].

Discussion and conclusion sections.  Fig 3 is absolutely unclear. Authors state that chemical reprogramming is a very powerful method, however I would recommend to exclude it completely from the text of the manuscript or change the aim of the paper – iPSC and iPSC derived cell therapeutic application- because I doubt that mouse cells will be clinically applied. 

Response: I understand the reviewer’s point because mouse cells have mainly been used in the mentioned studies although they reported the results of human cells [refs. 4, 175, 176]. This method should be examined further in a future using the human counterparts like organoids and organ tissues counterparts. However, it might take a long time to apply this technology to human clinical translational studies. Thus, we deleted this part of Figure 3, as suggested by the reviewer.

Reviewer 2 Report

Comments and Suggestions for Authors

This review by Lin and his team emphasizes the recent advancements in mitigating the tumorigenic risk associated with iPSC technologies. The review is comprehensive and includes numerous studies. However, the authors could consider revising certain aspects to make the review less detailed and more reader friendly.

Introduction

·   Authors could present the main factors that are used to reprogram cells (OSKM, Nanog)

·   Authors could succinctly cite the in vivo studies that have been conducted on the 4F mice (this review can help Cipriano 2023, doi.org/10.1038/s43587-023-00539-2 ).

Main text:

These three paragraphs:

·       Removal of epigenetic barriers in CSCs

·       Identification and lineage tracing of CSCs to reduce the cancer-initiation risk

·       Cancer risk prevention studies using cancer organoids

should be succinctly presented, repositioned, or omitted to maintain the review's primary focus. Their current length detracts from the main objective.

Minor points:

·       To make the review easier to read, authors should put all abbreviations of the gene/proteins

·       Line 125

·       Line 138: "For example, iPSCs were directly generated from mouse somatic cells using a cocktail of chemicals" which ones?

·       Line 317: “eraly stage”

·       Line 553 is in bold and line 560 is big.

·       Line 667: Same sentence twice.

Author Response

[Reviewer 2]

This review by Lin and his team emphasizes the recent advancements in mitigating the tumorigenic risk associated with iPSC technologies. The review is comprehensive and includes numerous studies. However, the authors could consider revising certain aspects to make the review less detailed and more reader friendly.

Response: I agree with this comment and revised the text as suggested by the reviewer.

Introduction:

  • Authors could present the main factors that are used to reprogram cells (OSKM, Nanog)

Response: As suggested by the reviewer, I added some text regarding this issue.

  • Authors could succinctly cite the in vivostudies that have been conducted on the 4F mice (this review can help Cipriano Nature Aging, 2023, doi.org/10.1038/s43587-023-00539-2 ).

Response: As suggested by the reviewer, I added in vivo studies to the review.

Main text:

These three paragraphs:

  • Removal of epigenetic barriers in CSCs
  • Identification and lineage tracing of CSCs to reduce the cancer-initiation risk
  • Cancer risk prevention studies using cancer organoids

should be succinctly presented, repositioned, or omitted to maintain the review's primary focus. Their current length detracts from the main objective.

Response: As suggested by the reviewer, I deleted these sections.

Minor points:

  • To make the review easier to read, authors should put all abbreviations of the gene/proteins

Response: As suggested by the reviewer, I added the abbreviation section (New Line 652-670).

  • Line 125 these vectors may themselves cause cancer by integrating into some specific or –

Response: I deleted this sentence.

  • Line 183: "For example, iPSCs were directly generated from mousesomatic cells using a cocktail of chemicals" which ones?

Response: I added the full information needed to address this issue.

  • Line 317: “eraly stage”

Response: I corrected this typing error.

  • Line 553 is in bold and line 560 is big.

Response: I corrected these errors.

  • Line 667: Same sentence twice.

Response: I corrected this error.